# Increased Enzyme Loading in PICsomes via Controlling Membrane Permeability Improves Enzyme Prodrug Cancer Therapy Outcome

**DOI:** 10.3390/polym15061368

**Published:** 2023-03-09

**Authors:** Akinori Goto, Yasutaka Anraku, Shigeto Fukushima, Akihiro Kishimura

**Affiliations:** 1Kyorin Pharmaceutical Co., Ltd., Watarase Research Center, 1848, Nogi, Nogi-machi, Shimotsuga-gun, Tochigi 329-0114, Japan; akinori.gotou@mb.kyorin-pharm.co.jp; 2Department of Bioengineering, Graduate School of Engineering, University of Tokyo, Hongo, Bunkyo-ku, Tokyo 113-8656, Japan; 3Innovation Center of NanoMedicne, Kawasaki Institute of Industrial Promotion, 3-25-14 Tonomachi, Kawasaki-ku, Kawasaki 210-0821, Japan; 4Department of Applied Chemistry, Faculty of Engineering, Kyushu University, 744 Moto-oka, Nishi-ku, Fukuoka 819-0395, Japan; 5Center for Molecular Systems, Kyushu University, 744 Moto-oka, Nishi-ku, Fukuoka 819-0395, Japan

**Keywords:** polyion complex, vesicle, protein loading, semi-permeable membrane, block copolymer, enzyme prodrug therapy, cancer therapy

## Abstract

Mesoscopic-sized polyion complex vesicles (PICsomes) with semi-permeable membranes are promising nanoreactors for enzyme prodrug therapy (EPT), mainly due to their ability to accommodate enzymes in their inner cavity. Increased loading efficacy and retained activity of enzymes in PICsomes are crucial for their practical application. Herein, a novel preparation method for enzyme-loaded PICsomes, the stepwise crosslinking (SWCL) method, was developed to achieve both high feed-to-loading enzyme efficiency and high enzymatic activity under in vivo conditions. Cytosine deaminase (CD), which catalyzes the conversion of the 5-fluorocytosine (5-FC) prodrug to cytotoxic 5-fluorouracil (5-FU), was loaded into PICsomes. The SWCL strategy enabled a substantial increase in CD encapsulation efficiency, up to ~44% of the feeding amount. CD-loaded PICsomes (CD@PICsomes) showed prolonged blood circulation to achieve appreciable tumor accumulation via enhanced permeability and retention effect. The combination of CD@PICsomes and 5-FC produced superior antitumor activity in a subcutaneous model of C26 murine colon adenocarcinoma, even at a lower dose than systemic 5-FU treatment, and showed significantly reduced adverse effects. These results reveal the feasibility of PICsome-based EPT as a novel, highly efficient, and safe cancer treatment modality.

## 1. Introduction

Polymeric vesicles have an inner aqueous cavity surrounded by a semi-permeable polymer membrane. Their therapeutic potential as enzyme-loaded vesicular nanoreactors has been increasingly recognized [1,2,3,4,5,6,7,8,9]. Specifically, vesicular nanoreactors with an approximate tuned size of 100 nm and a palisade of hydrophilic polymer strands, such as polyethylene glycol (PEG), can be used for targeted enzyme therapy of cancer via the systemic route. They accumulate in cancerous tissues through the enhanced permeability and retention (EPR) effect and avoid non-specific uptake by the reticuloendothelial system of the liver and spleen (i.e., stealth effect) [1,10,11]. Cancer-targeted enzymes in these nanoreactors selectively convert non-toxic prodrugs to their active form locally, killing cancer cells without causing non-specific systemic toxicity. This treatment method is termed directed enzyme prodrug therapy (DEPT). The use of semi-permeable polymeric vesicles is expected to increase the efficacy of DEPT because of their prolonged blood circulation and EPR of loaded enzymes in the tumor microenvironment (TME). Furthermore, the semi-permeable nature of the vesicular membrane allows for substrate exchange. Protection from the harsh exterior conditions of the TME can prolong the activity of the loaded enzymes.

Recently, we developed a novel DEPT system using polyion complex vesicles (PICsomes) composed of combinations of oppositely charged block/block or block/homopolymers [3,4,5,12]. PICsomes can be obtained as 100 nm-sized vesicles. They encapsulate enzymes of interest in their inner aqueous phase. Enzyme-loaded PICsomes show prolonged blood circulation after chemical crosslinking of their PIC membranes. Thus, they can effectively accumulate in tumor tissue via the EPR effect [4,5]. There are two unique features of PICsomes for the DEPT. The first is simple enzyme loading, wherein various enzymes can be loaded in PICsomes just by vortex mixing in an aqueous medium before the crosslinking treatment. The second is the modulable permeability of the vesicular membrane through precise tuning of its crosslinking degree. This unique property allows the import of an enzyme substrate from the exterior and the export of products to the outer space while holding the enzyme in its inner space. Moreover, the PIC membrane can be crosslinked to enhance its robustness, which can contribute to the preserved activity of the loaded enzymes in vivo. We successfully demonstrated that β-galactosidase-loaded PICsomes accumulated in a solid tumor following systemic administration. The administered substrate molecule was converted to the fluorescent product in the tumor over 4 days, which enabled in vivo fluorescence imaging [4]. These results implicated enzyme-loaded PICsomes as promising nanocarriers for DEPT.

Although non-crosslinked PICsomes in the solution can encapsulate enzymes upon vortexing, the lower efficiency of enzyme loading into PICsomes (the ratio of the loaded amount of enzyme to the total feed amount of enzyme) remains a significant hurdle for their practical application (e.g., 0.9% for β-galactosidase) [12]. To improve feed-to-loading efficiency, PICsomes need to be prepared at higher polymer concentrations. However, the size of PICsomes is affected by the polymer concentration during the preparation process; therefore, the polymer concentration needs to be fixed to obtain PICsomes of a specific size [10]. Another issue is the small number of enzyme molecules that can be loaded into the inner cavity of a PICsome. We previously reported encapsulation of only one or two molecules of β-galactosidase in a single PICsome [4,12]. For practical utilization of PICsomes as therapeutic nanoreactors, the amount of loadable target enzyme needs to be increased. One plausible idea is to increase enzyme concentration during the loading process. However, this would hamper the self-assembly of a uniform PICsome structure because of the increased viscosity and ionic strength of the solution [3,10,13,14,15]. In the present study, to overcome the low feed-to-loading efficiency of an enzyme and the small number of entrapped enzyme molecules in a PICsome, we developed a new method based on stepwise crosslinking (SWCL) of the PIC membrane for enzyme loading. In the SWCL method, the first crosslinking is used to enhance the robustness of PICsomes during the loading process at high enzyme/charged polymer concentrations and high ionic strength conditions. The second crosslinking is performed to confine the loaded enzymes in the inner cavity without leakage, including the in vivo environment, thereby preserving the proper semi-permeable nature of the PICsome membranes.

As previously demonstrated, cytosine deaminase (CD)-loaded PICsomes prepared via the SWCL method successfully converted the non-toxic prodrug 5-fluorocytosine (5-FC) to its active form, 5-fluorouracil (5-FU) [16]. Notably, 5-FU is a widely used anticancer drug. However, there are strong side effects, mainly due to its non-specific distribution in healthy tissues [16,17,18,19]. Alternatively, the 5-FC CD-convertible prodrug has a longer plasma half-life, lower toxicity in humans, and low cellular uptake in human cells [19]. These attributes make 5-FC a good candidate for prodrug. Moreover, the molecular weight of CD (~18 kDa) is considered to be suitable for the SWCL method according to the previous report [14], and CD could permeate the PIC membrane after the first crosslinking but not after the second crosslinking. Herein, we designed and developed CD-loaded PICsomes for the CD/5-FC DEPT system (Figure 1). Finally, the superior therapeutic effect and reduction of side effects were confirmed in the treatment of the tumor-bearing mice model compared to the effects of systemic administration of 5-FU alone.

## 2. Materials and Methods

### 2.1. Materials

Poly(5-aminopentyl-α,β-aspartamide) (P(Asp-AP)) with a degree of polymerization (DP) of 82 and poly(ethylene glycol) (PEG)-block-poly(α,β-aspartic acid) (PEG-PAsp, DP of PAsp = 75; Mn of PEG = 2000, Mw/Mn of PEG = 1.05) were prepared as reported previously [20]. Notably, 1-Ethyl-3-(3-dimethylaminopropyl) carbodiimide hydrochloride (EDC, Dojindo Molecular Technologies, Inc., Kumamoto, Japan), sulfo-Cy5 NHS ester (GE Healthcare Japan, Tokyo, Japan), Trypsin (Thermo Fisher Scientific Inc., Tokyo, Japan; used at 2.5%), 5-FC (Sigma-Aldrich Co. LLC., St. Louis, MO, USA), 5-FU (Tokyo Chemical Industry Co., Ltd., Tokyo, Japan), and passive lysis buffer (Promega Co., Tokyo, Japan) were used as received. CD from yeast in 80% ammonium sulfate solution (Calzyme Laboratories, Inc., San Luis Obispo, CA, USA) was used as received for all experiments, except to evaluate the interaction between the PIC membrane and CD. We prepared CD from yeast in 300 mM NaCl, 50 mM Tris-HCl, pH 8.0, and 10 mM dithiothreitol (DTT) to evaluate the interaction between the PIC membrane and CD.

### 2.2. Protein Expression and Purification

The open reading frame of the CD gene was codon-optimized for expression in *Escherichia coli* and subcloned into the *Nde*I and *Xho*I sites of pET-15b (GenScript, Piscataway, NJ, USA). His-tagged CD (His-CD) was expressed in *E. coli* BL21 (DE3)-RIL (Agilent, San Diego, CA, USA) with 0.5 mM isopropyl β-d-1-thiogalactopyranoside (IPTG) in LB Miller medium supplemented with 50 µg/mL ampicillin and 0.5 mM zinc-acetate at 16 °C for 16 h. Harvested bacteria were lysed in 50 mL lysis buffer (150 mM NaCl, 20 mM Tris-HCl, pH 8.0, 20 mM imidazole) by sonication using a UD-201 ultrasonic disrupter (Tomy Seiko Co., Ltd., Tokyo, Japan) at an output of 7 and 40% duty for three 5-min applications. The soluble lysate was fractionated by centrifugation at 16,000 rpm at 4 °C for 20 min and applied to a nickel-nitrilotriacetic acid (Ni-NTA) column containing 10 mL Ni-NTA superflow (QIAGEN, Valencia, CA, USA) equilibrated with lysis buffer. The column was washed with 100 mL wash buffer (150 mM NaCl, 20 mM Tris-HCl, pH 8.0, and 50 mM imidazole). Bound proteins were eluted with 100 mL elution buffer (250 mM NaCl, 20 mM Tris-HCl, pH 8.0, and 300 mM imidazole). His-tagged proteins were cleaved by thrombin using dialysis in dialysis buffer (250 mM NaCl, 20 mM Tris-HCl, pH 8.0) at room temperature for 3 h. Precipitation was observed during the dialysis. The precipitate was removed by centrifugation at 16,000 rpm for 30 min at 4 °C. The supernatant was applied to HiTrap Benzamidine FF (Cytiva, Marlborough, MA, USA) and Ni-NTA Superflow QIAGEN) columns, 1 and 2 mL, respectively. The flow-through was collected. After concentration to 20 mL by ultrafiltration with AmiconUltra (molecular weight cut-off [MWCO] 30,000; Merck, Kenilworth, NJ, USA), proteins were fractionated by size-exclusion chromatography (SEC) in SEC buffer (25 mM Tris-HCl, pH 8.0, 300 mM NaCl, 10 mM DTT) with Superdex 200 HiLoad 16-600 (Cytiva) at 1.5 mL/min using a BioLogic Duoflow FPLC system (Bio-Rad, Hercules, CA, USA). The final product was buffer-exchanged using ultrafiltration and concentrated to 10 mg/mL. The protein concentration was determined by the ultraviolet method using an extinction coefficient of 1.12 at 280 nm.

### 2.3. Material Characterization

Dynamic light scattering (DLS) measurements were performed on a Zetasizer-nano (Malvern Instruments Ltd., Worcestershire, UK) to determine the particle size distribution. Transmission electron microscopy (TEM) was performed using a JEM-1400 microscope (JEOL Ltd., Tokyo, Japan) operating at 120 kV. The samples for TEM were dispersed in purified water and transferred onto 400 mesh size copper grids coated with a thin film of collodion and carbon, followed by staining with uranyl acetate. Fluorescence intensity was measured using a NanoDrop 3300 spectrophotometer (Thermo Fisher Scientific, Waltham, MA, USA).

### 2.4. Cell Lines and Animals

A549 human lung adenocarcinoma epithelial cells (National Cancer Center, Tokyo, Japan) and C26 murine colon adenocarcinoma cells (CRL-2638, Alexandria Technical & Community College, Alexandria, MN, USA) were cultured in RPMI-1640 (Sigma-Aldrich) supplemented with 100 U/mL penicillin (Sigma-Aldrich), 100 µg/mL streptomycin (Sigma-Aldrich), and 10% fetal bovine serum (FBS) (Thermo Fisher Scientific). Female, 5-week-old BALB/c mice (Oriental Yeast, Tokyo, Japan) were used for in vivo experiments.

### 2.5. Preparation of CD@PICsomes

P(Asp-AP) and PEG-PAsp were separately dissolved in 10 mM phosphate buffer (PB, pH 7.4) at a final concentration of 1.0 mg/mL. Typically, PEG-PAsp solution (400 μL) was mixed with 530 μL of P(Asp-AP) solution to maintain an equal unit molar ratio of –COO^−^ and –NH_3_^+^ in the mixed solution. The solution was mixed by vortexing at 2000 rpm for 2 min to form PICsomes. For the first crosslinking, the reaction mixture was treated with 540 μL of 1.0 mg/mL EDC solution, which contained equal amount of EDC with the amount of –COO^−^ in PAsp, left statically overnight at room temperature. The resulting solution was purified using a polyethersulfone (PES) ultrafiltration membrane (MWCO = 300,000) at 25 °C. The buffer exchange was repeated 10 times to completely remove byproducts from the reaction mixture, resulting in pre-crosslinked empty PICsomes (Pre-CL PICsomes). The final polymer concentration was adjusted to 10 mg/mL (10-fold dilution). The Pre-CL PICsomes were blended with 10 mg/mL CD solution (80% saturated ammonium sulfate solution) at a 5:8 volume ratio and then vortexed at 2000 rpm for 2 min. The mixture was treated with 540 μL of 50 mg/mL EDC solution for the second-step crosslinking and incubated overnight at 4 °C. The resulting solution was purified using a PES ultrafiltration membrane (MWCO = 300,000) at 5 °C. The buffer exchange was repeated 10 times to remove byproducts and free CD from the solution, to yield CD-loaded PICsome (CD@PICsome).

To optimize the crosslinking conditions of Pre-CL PICsomes, four different EDC concentrations were tested (1, 3, and 5 mg/mL). Pre-CL PICsomes were prepared using the same protocol as that of CD@PICsome, except for EDC concentrations. For reference CD loading experiments, the solutions of P(Asp-AP), PEG-PAsp, and CD (final CD concentration: 0.1–0.3 mg/mL) were mixed, or the P(Asp-AP) solution was mixed with the PEG-PAsp solution and then mixed with the CD solution (final CD concentration: 0.1–0.3 mg/mL).

### 2.6. Preparation of Fluorescence Modified CD-Loaded PICsomes

To calculate the polymer concentration, Cy5-labeled Pre-CL PICsomes (Cy5-Pre-CL PICsomes) and CD@PICsomes with Cy5-labeling (CD@Cy5-PICsome) were prepared using Cy5-conjugated PEG-PAsp (PEG-PAsp-Cy5) instead of non-labeled PEG-PAsp, as previously reported [20]. To investigate the feed-to-loading efficiency (fLE) and leakage amount of CD, Cy5-labeled CD (Cy5-CD) was prepared from sulfo-Cy5 NHS ester and CD in 10 mM PB. The reaction mixture was purified by ultrafiltration using a PES membrane (MWCO = 3000) to obtain Cy5-CD. The loading of Cy5-CD into PICsomes was performed using the same protocol as that of CD@PICsome, and four different concentrations of Cy5-CD were examined (1.0, 2.5, 5.0, and 10 mg/mL). To test the interaction between the PIC membrane and CD, Cy5-CD-loaded PICsomes (Cy5-CD@PICsomes) were prepared. 

The PEG-PAsp-Cy5 concentration of PICsomes, Cy5-CD loading efficiency, and leakage amount of Cy5-CD from PICsomes were evaluated using a gel permeation chromatography (GPC) system equipped with a fluorescent detector (excitation/emission = 643/667 nm) and a Superdex 200 column (GE Healthcare, Chicago, IL, USA). As a mobile phase, 10 mM of phosphate-buffered saline (PBS, pH 7.4) was used. 

### 2.7. Interaction of Pre-CL PICsomes and CD

Enzyme activity was calculated from the conversion of 5-FC to 5-FU. The amounts of 5-FU and 5-FC were determined using a high-performance liquid chromatography system equipped with an ultraviolet detector (wavelength: 270 nm) and an ODS-3 C18 column (GL Sciences, Shanghai, China). A 19:1 mixture of 10 mM potassium dihydrogen phosphate and acetonitrile was used as the mobile phase. As a reference, CL Cy5-PICsomes were prepared as previously described [10]. To measure enzyme activity, CD@PICsomes were dispersed in 10 mM PBS (pH 7.4) to obtain a total polymer concentration of 14 mg/mL. A 20 μL of diluted CD@PICsome dispersion was mixed with 1.48 mL of 0.01 mg/mL 5-FC and incubated at 40 °C. The conversion of 5-FC to 5-FU was determined at 10, 20, 30, 40, 50, and 60 min after mixing. To investigate the effect of CD concentration on the enzymatic activity of CD-loaded PICsomes, CD@PICsomes were prepared using four different CD concentrations (1.0, 2.5, 5.0, and 10 mg/mL). Then, each enzyme activity of obtained samples was measured as shown above.

For further investigation, CD@PICsome were diluted to 0.2 mg/mL of polymer concentration. The diluted sample was mixed with 0.25% trypsin at a volume ratio of 1:1. The mixture was incubated at 37 °C for 5 min. Then, enzyme activity of CD@PICsome treated with trypsin was measured by above method.

### 2.8. Evaluation of Stability of CD@PICsome Based on Enzyme Activity Assay

To determine the polymer concentration, CD@Cy5-PICsome was used. To measure enzyme activity, the CD solution was diluted 100-fold with 10 mM PBS (150 mM NaCl, pH 7.4) to a final CD concentration of ~0.1 mg/mL. Enzyme activity of CD@PICsome was measured by the method described in 2.7. To confirm the stability of CD over the period used in the in vivo experiment (vide infra), the enzyme activities of CD@PICsome and CD were measured at 8, 16, 80, and 152 h after preparation (*n* = 3; initial enzyme activity adjusted to 0.47 U/mL at 40 °C for both cases). For the confirmation of durability against repeated use, the enzyme activity was measured for CD@PICsome at 40 °C (up to 10 runs for each CD@Cy5-PICsome; *n* = 3).

### 2.9. Evaluation of In Vitro Cytotoxicity

The cytotoxicity of the 5-FC and CD@PICsome combination was evaluated using A549 and C26 cells. Each cell line was seeded onto a 96-well plate (2.5 × 10^4^ cells/mL) and pre-incubated with 90 µL of the culture medium for 24 h at 37 °C under a humidified atmosphere containing 5% CO_2_. Subsequently, 5 µL of 0.2 U/mL CD@PICsome was added, followed by the addition of 5 µL of 0.04 or 0.08 mg/mL of 5-FC. As reference experiments, cells were treated with 5-FC (Table 1, samples 2 and 3), CD@PICsome (sample 1), or 5-FU (samples 6 and 7) at the same concentration as samples 4 or 5; 10 mM PBS (pH 7.4) was used as a negative control (sample 8). All samples were incubated in multi-well plates at 37 °C in a humidified atmosphere containing 5% CO_2_. After a 48 h or 72 h incubation, viability was measured using a cell counting kit-8 (Dojindo Molecular Technologies, Inc., Rockville, MD, USA) by incubating for 1 h at 37 °C in a humidified atmosphere containing 5% CO_2_. The absorbance of each well was measured at 450 nm using a model AD200 microplate instrument AD200 (Beckman Coulter Inc., Tokyo, Japan) (*n* = 3). Cell viability was calculated using the following formula:Cell viability (%) = {[(*A*_SAM_) − (*A*_BLK_)]/[(*A*_PBS_) − (*A*_BLK_)]} × 100(1)
where *A*_SAM_ is the absorbance of samples 1–7 in Table 1, *A*_PBS_ is the absorbance of the control PBS (sample 8), and *A*_BLK_ is the absorbance of the blank.

### 2.10. Evaluation of Plasma Clearance and Biodistribution and Hepatotoxicity of CD@PICsome

Each BALB/c mouse (*n* = 3) was subcutaneously inoculated with 100 µL of suspension of C26 cells in RPMI-1640 containing 10% FBS (1.0 × 10^6^ cells/mL). The tumors were allowed to grow for 10 days. First, 200 μL of Cy5-CD@PICsome, which corresponded to 0.5 U/mL in 10 mM PBS (pH 7.4), was administered by intravenous (i.v.) injection through the tail vein of each mouse. At 24 and 72 h after injection, blood was collected from the postcaval vein using heparinized syringes, and the main organs (lung, kidney, liver, and spleen) and C26 tumor tissue were excised. After centrifugation of the collected blood at 20,000× *g* for 5 min at 4 °C, the fluorescence of the plasma was measured using an Infinite M1000 pro microplate reader (Tecan, Männedorf, Switzerland). The main organs were homogenized with 5× passive lysis buffer (Promega KK., Tokyo, Japan). The resulting colloidal solutions were measured using a microplate reader (excitation and emission wavelengths of 643 and 667 nm, respectively). 

To evaluate hepatotoxicity, C26-bearing mice were prepared (*n* = 3) and tumors were allowed to grow for 7 days. On day 1, 200 μL of 0.5 U/mL CD@PICsome in 10 mM PBS (pH 7.4) was administered by i.v. injection through the tail vein of each mouse. Then, 200 µL of 5-FC solution (10 mM PBS, pH 7.4; 5-FC concentration: 80 mg/kg) was similarly administered on days 4, 8, 11, and 15 (group 2 of Table 2). For comparison, mice were treated with CD@PICsome without further administration of 5-FC, 5-FU (80 mg/kg), 5-FC (80 mg/kg), or 10 mM PBS (pH 7.4) following the administration schedule shown in Table 2. Furthermore, 200 µL of blood was collected from the blood vessel under the chin using heparinized syringes 1 h after the injection of 5-FC, 5-FU, or 10 mM PBS. Plasma was collected by centrifugation at 20,000× *g* for 5 min. Aspartate aminotransferase (AST), alanine aminotransferase (ALT), and alkaline phosphatase (ALP) levels were measured using a Dri-Chem 7000z (FUJIFILM Corporation, Tokyo, Japan).

### 2.11. Evaluation of In Vivo Therapeutic Effect

To evaluate in vivo therapeutic effect, C26-bearing mice were prepared as described above (*n* = 5). Tumors were allowed to grow for 7 days. Further, 200 μL of 0.5 U/mL CD@PICsome in 10 mM PBS (pH 7.4) was administered by i.v. injection through the tail vein of each mouse on day 1. Subsequently, 200 µL of 5-FC solution (10 mM PBS, pH 7.4) was administered by i.v. injection on days 4, 8, 11, and 15 (Table 2, groups 1 and 2). As a positive control, 5-FU was similarly injected without the administration of CD@PICsome (Table 2, groups 3 and 4). As negative controls, mice were treated with CD@PICsome (Table 2, group 5), 5-FC (groups 6 and 7), and 10 mM PBS (pH 7.4; group 8). Two different concentrations (10 and 80 mg/kg) were administered. Mouse body weight and tumor size were measured twice a week (on days 1, 4, 8, 11, 15, 18, 22, 25, and 29). The tumor size was determined using a digital Vernier caliper across its two perpendicular diameters. Tumor volume (V) in mm^3^ was calculated as the following:*V* = (*a*^2^ × *b*)/2(2)
where *a* and *b* are the minor and major diameter of the tumor, respectively.

Mouse body weight and tumor size were monitored until the tumor size was >2500 mm^3^ and weight loss was >25%. Mice that met either of the criteria were sacrificed. The experiment was completed when three mice died or were euthanized in the group.

### 2.12. Statistical Analysis

The calculation of mean value and standard deviation in figures, regression analysis, and Student’s *t*-test were conducted using Excel software, version 2011 (Microsoft, Washington, DC, USA). For the *t*-test performed in cytotoxicity and therapeutic assays, a *p*-value < 0.05 was considered statistically significant.

## 3. Results and Discussion

### 3.1. Preparation of Crosslinked PICsomes Loaded with CD (CD@PICsomes) by SWCL

Increasing the loading amount and feed-to-loading efficiency (fLE) of enzymes into PICsomes is key to their practical use in DEPT. The minimum fLE of enzymes in PICsomes using the conventional method of vortex mixing was limited to <1% [4]. In the conventional method, enzyme molecules and pre-assembled empty PICsomes are mixed, followed by vortex mixing to disassemble the PICsomes into unit PICs. The PIC structure comprises polyanions and polycations with minimal association to compensate for their charge [21]. After the vortex mixing is stopped, PICsomes spontaneously regenerate through the self-assembly of unit PICs. This regeneration allows the entrapment of enzymes that are in the solution [4]. The process of enzyme entrapment proceeds stochastically, which results in not enough loading efficiency to apply to DEPT. The efficiency can be improved by increasing the concentration of the polymer and enzyme in the solution. However, the increased polymer concentration leads to the formation of larger PICsomes, which is unfavorable for EPR-mediated tumor accumulation [10]. In the present study, the conventional method did not work properly for CD loading at higher enzyme concentrations in the polymer system (P(Asp-AP)/PEG-PAsp), resulting in the formation of micrometer-sized aggregates. Presumably, the high salt concentration of the CD stock solution (80% saturated ammonium sulfate solution; final concentration of ammonium salt is approximately 0.5 g/mL) may hamper the stable polyion pairing in the mixed enzyme–polymer solution. In fact, under higher salt concentration conditions, robust polyion pairing is required to keep the unilamellar structure and well-controlled size of the PICsome [3,15]. These issues associated with CD loading into P(Asp-AP)/PEG-PAsp PICsomes motivated us to develop the SWCL method.

The SWCL method comprises two steps. The first step involves loose crosslinking to maintain the unilamellar PICsomes with a well-controlled size against environmental stresses (viscosity and ionic strength) [21]. The second step involves crosslinking that stably confines the loaded enzymes into the inner cavity while maintaining the proper semi-permeable nature of the PICsome membranes. As previously reported, PIC membranes with a lower degree of crosslinking can have higher permeability, which allows the permeation of macromolecules, such as linear PEG with a molecular weight (M.W.) of 42,000 and branched PEGs with an M.W. of 10,000. In the case of branched PEGs with a hydrodynamic radius of 2.5–2.7 nm, which were loaded into PICsomes with lower crosslinking, approximately 15% of release occurred over 1 week [14]. Therefore, CD with an approximate M.W. of 18,000 (2.0 nm calculated hydrodynamic radius [22]) would permeate the PICsome membrane. The findings indicated that Pre-CL PICsomes are favorable for enzyme loading. In addition, to enhance the quantity of CD loaded into Pre-CL PICsomes, loading could be performed in the presence of a high concentration of PICsomes, enzymes, and salt. Pre-CL PICsomes are considered to be stable in these harsh conditions. Moreover, the second crosslinking suppressed the leakage of CD from the PICsomes. Thus, the SWCL method appears effective for loading and maintaining enzymes inside PICsomes.

First, Pre-CL PICsomes were prepared with an amount of EDC equal to the carboxyl groups in the polymer. This crosslinking treatment was identical to the method we previously reported, in which PICsomes with a crosslinking degree of 45% that were obtained displayed a permeation of linear PEG with an M.W. of 42,000, and branched PEGs with an M.W. of 10,000 [14]. The second crosslinking treatment was performed using a 50-fold EDC to the original amount of carboxyl groups in the polymer, in which the crosslinking degree of PICsomes was estimated to be ~90% [14]. Such a higher crosslinking degree would contribute to the prevention of leakage of loaded CDs. The fLE of CD@PICsomes was confirmed by GPC using Cy5-CD@PICsomes without purification. The fLE (%) was calculated as follows:([Peak area of (1)]/[Sum of peak area of (1) and (2)]) × 100(3)

The fLE of CD@PICsome was approximately 9% 1 h after the addition of the CD solution (GPC data not shown) and 19% after 24 h (GPC data not shown). This fLE was considerably higher than that reported for previous enzyme-loaded PICsomes (fLE < 1%) [4,5,12]. However, this loading method is time-consuming and may cause the deactivation of the enzyme. Therefore, we decided to use vortex mixing to facilitate enzyme loading. Surprisingly, two minutes of vortex mixing for CD loading enhanced the fLE to 44% (Figure 1A and Appendix A). Peak (1), which was eluted at the same elution time as Cy5-labeled empty PICsomes, is assignable to Cy5-CD@PICsome. Peak (2) was assigned to a free Cy5-CD. The PICsomes prepared by the SWCL method showed successful encapsulation and retention of the enzyme. Notably, the CD content of the CD@PICsome formulation was estimated to be ~41 *w*/*w*% based on the fLE value. Thus, enzyme activity-based loading efficiency (LE) was calculated from the enzyme activity measurements. We estimated the enzyme activity using “Unit (U)” as an indicator (1 U is defined as the amount of enzyme that converts 1 μmol of substrate per minute at 40 °C). The enzyme activity of the obtained CD@PICsome was 16 U, while the enzyme activity of the CD used for loading was 60 U, indicating that 27% enzyme activity was maintained. Considering the fLE of 44%, ~40% of enzyme activity was lost, probably due to deactivation by the chemical influence of EDC and/or physical impact imposed during the encapsulation and purification processes. It is noteworthy that DLS measurements revealed no significant difference in the average size and polydispersity index (PDI) before (size = 101 nm, PDI = 0.06) and after (size = 105 nm, PDI = 0.08) CD loading treatment via vortex mixing. In addition, no significant difference was observed in the morphology of the particles before and after the CD loading treatment (Figure 1B). Thus, the vortex-based SWCL method was helpful for the enhancement of fLE and enzyme content without deterioration, and in further experiments, we decided to use the vortex-based method.

Next, to confirm the effect of crosslinking of the Pre-CL PICsomes on CD loading, CD was loaded into Pre-CL PICsomes prepared using 3 and 5 mg/mL of EDC, respectively. Previously, the crosslinking degrees of Pre-CL PICsome were reported to be 61% (using 3 mg/mL of EDC) and 77% (using 5 mg/mL of EDC) [14]. The fLEs of CD@PICsome were 1.4% (using 3 mg/mL of EDC) and 1.2% (using 5 mg/mL of EDC), using 2 min of vortex mixing treatment. Only a small amount of CD was loaded onto CD@PICsome. These results suggest that a higher crosslinking degree of the Pre-CL PIC membrane makes the resulting polymer network denser, which disturbs the effective loading of CD. 

To confirm the effect of the second crosslinking on the long-term use and storage of the CD@PICsome, we tested the leaching of Cy5-CD from purified Cy5-CD@PICsome after the second crosslinking using GPC. No leaching of Cy5-CD was observed for Cy5-CD@PICsome 7 and 14 days after purification when treated with a large excess amount of EDC for the second crosslinking (Appendix A; 9.0 eqs. or more for the –COO^−^ groups of the original PEG-PAsp). However, after treatment with a much lower amount of EDC for the second crosslinking (1.0, 3.0, and 5.0 eqs. to the –COO^−^ groups), Cy5-CD@PICsomes showed a time-dependent release of Cy5-CD, the behavior of which varied with the amount of EDC used (Appendix A). Thus, CD@PICsomes treated with an excess amount of EDC for the second crosslinking were considered to be stable enough for further investigation. Therefore, 50 mg/mL of EDC was used for the following experiments.

Collectively, the SWCL method provides more options for enzyme loading conditions by improving PICsome stability via the first crosslinking step and suppressing payload leaching by the second crosslinking step. However, the SWCL method is limited in that it does not allow the loading of enzymes exceeding a certain M.W. Concerning enzymes previously used as PICsome cargoes [4,5,12], the SWCL method is suitable for lysozyme (~14,000) and asparaginase (~40,000), but not β-gal (~540,000).

### 3.2. Interaction of Pre-CL PICsomes and CD

The fLE value found for the SWCL method was >50 times higher with vortex mixing than the estimated value of 0.8% (according to volume fraction, the details of estimation are described in the Appendix A). Therefore, the contribution from the mechanical impact of the vortex mixing is limited (~2.3-fold increase for the SWCL without vortex mixing), and it is reasonable to assume that the condensation of CD occurred in the PICsome compartment regardless of the mechanical impact from the vortex mixing. One possible explanation is the interaction between CD and the PIC membrane. Thus, treatment of CD@PICsome with trypsin was performed to determine the location of CD. Trypsin treatment resulted in a loss of approximately 70% of the enzyme activity. The M.W. of trypsin (23,000) rendered it incapable of permeating the PIC membrane after the second crosslinking. Therefore, ~70% of the loaded CD would interact with the PIC membrane, which contributed to the inactivation of CD by trypsin.

To gain more insight into the interaction of CD with Pre-CL PICsomes, the effect of CD concentration on fLE during the CD loading process was investigated using Cy5-CD. The CD concentration and fLE showed a good linear relationship (Figure 2A, GPC chromatograms are shown in Appendix A). This result suggests that a higher concentration of CD accelerates the encapsulation process (Figure 2B). One possible explanation for the accumulation of CD in the membrane is the electrostatic interaction and/or hydrophobic interaction. In addition, the PIC membrane may show an upper limit for the accommodation of CD because of its limited space, with a plateau in the amount of CD in the membrane. However, such an upper limit was not found in this CD concentration range. This finding indicates that only a rising part of the curve was observed in the range of this experiment (Figure 2B). The fLE value obtained in this section (29% for 10 mg/mL CD solution) was lower than the values in the previous section (44%). Plausible factors include different sources of CD (in this section, we prepared CD, while for the experiment in the previous section, we used a commercially available CD), and the difference in solution composition (our product was dispersed in 25 mM Tris-HCl, pH 8.0, 300 mM NaCl, 10 mM DTT; the commercial product was dispersed in 80% saturated ammonium sulfate solution).

Next, the effect of the feed concentration of CD on the enzyme activity of CD@PICsomes was investigated using CD without Cy5 labeling (Figure 3). Negligible enzyme activity was observed for the products obtained at a feed CD concentration of ≤2.5 mg/mL, whereas much higher activity was confirmed in the case of a feed CD concentration >2.5 mg/mL. A possible explanation for this difference is the inactivation of CD by the EDC treatment used for the second crosslinking. In fact, the EDC used for the second crosslinking was in excess of the total number of carboxylate groups in the PICsome membrane. Therefore, it is conceivable that the residual EDC reacted with CD, resulting in its deactivation. This effect would be more remarkable at lower CD concentrations. 

The average particle size of the Pre-CL PICsomes used in this section was 135 nm, and the average particle size of CD@PICsome was 147 nm. Assuming a PIC membrane thickness of 15 nm [11], the volume expansion rate of the PIC membrane was 121%. According to the fLE value of 29% for 10 mg/mL CD (Figure 2), 4.6 mg of CD was loaded per 10 mg of polymer, and 3.2 mg of CD was assumed to be in the membrane, based on the 70% activity loss by trypsin treatment. The remaining 30% of CD would be out of the trypsin permeable area. 

### 3.3. Stability of CD@PICsome Based on Enzyme Activity

Next, the enzyme activity change of CD@PICsome was tested in 10 mM PBS at 40 °C, which is close to the body temperature of a mouse (Figure 4A). The free CD was completely deactivated within 72 h, whereas CD@PICsome maintained 50% of the initial enzyme activity even after 150 h. The enzyme activity change of CD@PICsome and 5-FU production for repetitive use were measured (Figure 4B,C). The enzyme activity of free CD decreased in proportion to the increase in run number and was completely deactivated after ≥6 runs. On the other hand, the enzyme activity of CD@PICsome was almost constant over 10 runs, and CD@PICsome lasted until the production of 5-FU. These results indicate that CD was stabilized by loading into PICsomes, which can contribute to prolonging enzyme activity lifetime and increasing durability. A confined environment provided by PICsomes may contribute to the protection of enzymes from deactivation [3,4,5,12]. Thus, CD@PICsome is expected to be more stable and would work continuously under in vivo conditions. Notably, CD stabilized by the protection of PICsomes leads to the possibility of storage under milder conditions and does not need to be stored at low temperatures.

### 3.4. Evaluation of Cytotoxicity of CD@PICsome

Before proceeding to the in vivo experiments, the in vitro cytotoxicity of CD@PICsome was examined (Figure 5). Cell viability was not affected by 5-FC or CD@PICsome. On the other hand, significant cytotoxicity was observed after co-treatment with CD@PICsome and 5-FC, comparable to that of the 5-FU treatment. Slightly lower cytotoxicity was observed for the CD@PICsome and 5-FC co-treatment after 72 h. This can be attributed to the gradual deactivation of CD, as shown in Figure 4A. Thus, cytotoxicity was successfully confirmed when 5-FC and CD@PICsome co-existed, and the enzyme nanocarriers and prodrugs were not harmful in terms of acute toxicity. Considering that 5-FC is not transported into the cytoplasm through the plasma membrane, its conversion to 5-FU by CD@PICsome likely occurred in the extracellular medium [23].

### 3.5. Evaluation of Plasma Clearance, Biodistribution, and Hepatotoxicity of CD@PICsome

The biodistribution of Cy5-CD@PICsome was investigated to gain more insight into the therapeutic application of CD@PICsome. Regarding the plasma clearance of Cy5-CD@PICsome, the residual fraction to the dose was 14.9% and 9.4% in blood at 24 and 72 h after injection, respectively. The order of organ uptake of Cy5-CD@PICsome was liver > C26 tumor ~ spleen >> lung and kidney (Figure 6). These findings indicated that accumulation of Cy5-CD@PICsome preferentially occurred in tumor tissues of major organs other than the liver. Similar prolonged blood circulation time [5,11,24] and tumor accumulation behavior were observed for previously reported PICsomes [11,24], indicating that CD@PICsome did not lose its pharmacokinetic advantage compared to previous PICsomes, even after loading a large amount of enzyme.

Based on the biodistribution results, CD@PICsome accumulation occurred most frequently in the liver. Therefore, hepatotoxicity after administration of CD@PICsome and 5-FC was investigated. The values of AST, ALT, and ALP for group 2 (CD@PICsome and 80 mg/kg 5-FC) were similar to those of the other control groups (Figure 7). These results clearly show that the EPT of CD@PICsome and 5-FC does not damage the liver and can be safely used for therapeutic purposes. The impermeability of 5-FC to the cell membrane may suppress its conversion from 5-FC to 5-FU in the liver.

### 3.6. Evaluation of In Vivo Therapeutic Effect of CD@PICsome

Next, CD@PICsome was used for in vivo experiments using tumor-bearing mice. In fact, PICsomes with a diameter of 100 nm showed selective accumulation in C26 tumor tissues due to the EPR effect [11]. We planned the experiment as follows (see Table 2 for detailed information). First, C26 tumor cells were subcutaneously inoculated as a model tumor. Then, CD@PICsome was administered on day 1. A 3-day period followed so that CD@PICsome could accumulate in the tumor tissue by the EPR effect and be eliminated from the bloodstream [4]. Finally, 5-FC was administered on days 4, 7, 11, and 14 at different concentrations (group 1, 10 mg/kg; group 2, 80 mg/kg). As a control, only PBS was administered (group 8; negative control).

To discuss some major trends clearly, selected results are displayed in Figure 8 (all individual data are shown in Appendix A). Remarkably, after CD@PICsome administration, significant tumor growth suppression was confirmed for the dose condition of 10 mg/kg of 5-FC from day 8 (Figure 8, group 1), although the therapeutic effect was hardly observed for the 5-FC and CD@PICsome administration groups (Figure 8, groups 5 and 6). These results suggested that the DEPT system worked well in vivo. Moreover, group 1 showed a superior therapeutic effect compared to group 3. In addition, an obvious delay in the onset of weight loss and significant suppression of weight loss were confirmed for group 1, whereas group 3 showed limited suppression of weight loss (Appendix A). This result can be mainly explained by the prolonged blood half-life of 5-FC (3–4 h [19]) compared to that of 5-FU (8–20 min [18]), which allows for the lasting conversion of 5-FC to 5-FU by CD@PICsome at the tumor site.

In the case of the 80 mg/kg dose, group 4 (treated with 5-FU) showed a remarkable antitumor effect. However, all mice died before day 15 (Figure 8). This side effect was more serious than that in group 3 (low-dose condition). Notably, group 2 (treated with CD@PICsome and 5-FC) showed apparent tumor growth suppression compared to the PBS group (Figure 8), and negligible body weight loss on any day despite tumor growth (Appendix A). This indicates a better prognosis after co-treatment with CD@PICsome and 5-FC. In particular, the antitumor effect of group 2 was comparable to that of group 4 until day 7; on the other hand, tumor growth was explicitly found after day 11, although the effect of tumor growth suppression was consistently better than that of group 8 after day 7. The difference in therapeutic efficacy between 5-FU treatment and 5-FC/CD@PICsome co-treatment is probably attributed to the gradual deactivation of CD after day 7. PICsome-based EPT can provide much safer treatment than treatment using 5-FU alone, that is, a higher therapeutic effect is expected for the lower dose of 5-FC and lower toxicity for the higher dose, although the dose of CD, the administration schedule, and dose of 5-FC should be optimized in the future.

## 4. Conclusions

The SWCL method allows for semi-permeability tuning of PIC nanomembranes and successful loading of yeast CD into the PICsomes with high loading efficiency and loading amount. The high efficiency of fLE may be due to the interaction between the CD and Pre-CL PICsomes. Furthermore, the effective action of EPT based on CD@PICsome and 5-FC was confirmed in both in vitro and in vivo conditions, and its therapeutic efficacy for tumor treatment was demonstrated through mouse models. Remarkably, EPT utilizing CD@PICsomes significantly suppressed tumor growth under low-dose conditions, while sole administration of 5-FU did not show therapeutic efficacy. In addition, the EPT utilizing CD@PICsomes showed negligible side effects under the high dose condition, in which the sole administration of 5-FU was lethal. Considering that 5-FC is more favorable for oral administration in terms of bioavailability [25,26], the EPT utilizing CD@PICsomes seems certainly promising due to the possibility of reducing patient burden. However, the administration schedule of both CD@PICsomes and 5-FC was not optimized at this stage. Further investigation is required to fully utilize the CD@PICsome/5-FC system. 

The SWCL method can be applied to other enzymes, which may expand the scope of EPT for the treatment of other diseases. However, CD may have a specific interaction with the PIC membrane, contributing to higher fLE and enzyme activity. It is necessary to verify whether such effective loading can be found for other enzymes. Furthermore, the SWCL method is potentially compatible with the co-loading of several different enzymes into a single PICsome, which can contribute to prodrug-combination therapy and novel prodrug designs based on a tandem enzyme reaction inside the PICsomes. Thus, PICsome-based EPT is potentially useful for the treatment of a wide variety of diseases, including cancer. This could open a new avenue in drug therapy.

## 5. Patents

JPWO2014133172A1, CN105188905B, US10471019B2, US10322092B2.

## Data Availability

All data are available upon reasonable request.

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
