# Peer review of "Increased Enzyme Loading in PICsomes via Controlling Membrane Permeability Improves Enzyme Prodrug Cancer Therapy Outcome"

_polymers, 2023, doi:10.3390/polym15061368_

Round 1

Reviewer 1 Report

Dear Authors, let me first congratulate you for this niece piece of work, which opens potential new avenues for cancer tehrapy. I understand that in vivo experiments are not easy to perform, but in the present version of your manuscript these data are to me the weakest.

First of all figure 8, 9 and 10 is represented in both panels (A) and (B), thus 5 times, the same control group 1, for which no indication whatsoever of the number of repeat and/or animals is indicated.

Please find a way to represent your data differently (single figure), to avoid misunderstanding for the reader. Also explicitly mention the number of repeats and animals used in each case, so that the referee and readers may appreciate the signification of your results.

It would also be important to show the actual tumor size in mm2 instead of the relative tumor volume that you are displaying in the Y axis.

Second, additional details regarding the source of the CD (Mouse / Human or other specie) should be given, this sentence is not informative enough  « ..The open reading frame of the CD gene was codon-optimized.. » -Please comment in the discussion whether the optimization or different specie might compromise or enhance the therapeutic effect (immune reactivity etc…) ?

Line 478 you mention that the mouse body temperature is 40°C ! I usually thought like many others that it would rather be around 36.9°C Please change

Figure 5 (A) and (B) could you please add below your graphic the actual treatments instead of numbers

You have been using the CT26 colorectal carcinoma mouse cell line, haven’t you ? If so please change in the text and provide its ATCC number.

Author Response

Dear reviewer 1,

 Thank you for your comments. Here, we describe the point-by-point responses to your comments and clarify the revised parts.

Reviewer 2 Report

The authors have developed nanoreactor polyion complex vesicles that selectively enter cancer cells and remain in the systemic circulation for a long time. The semipermeable shell of the vesicles in the cancer cell allows the entry of the prodrug into the aqueous core of the nanoreactor, where the enzyme converts the prodrug into an active form, which then leaves the interior of the polyion complex vesicles (i.e., the nanoreactor) and exerts a pharmacological effect on the cancer cell.

The authors examined the nanoreactor vesicles in detail using physico-chemical methods, as well as the interaction with cancer cells using biopharmaceutical methods.

These researches are very current and lucid, a nanoreactor (a small chemical laboratory) is selectively introduced into the cancer cell, where the prodrug is then transformed into an active form.

Minor:

1. The size of PICsomes is affected by the polymer concentration during the preparation process; therefore, the polymer concentration needs to be fixed to obtain PICsomes of a specific size. How the length of the polymer chain affects the size of polyion complex vesicles, as well as their stability. What is the polydispersity (by size) of the applied polymer.

2. The authors could explain in more detail the selectivity of penetration of polyion complex vesicles towards cancer cells, i.e., their reduced entry into the endoreticular system of the liver and spleen.

3. Based on the size of the inner cavity of the polyion complex vesicles and the size of the tested enzyme, which is the maximum number of enzymes that can be placed in the cavity of the nanoreactor (at an extremely high enzyme concentration).

Author Response

Dear reviewer 2,

 Thank you for your comments. Here, we describe the point-by-point responses to your comments.

Round 2

Reviewer 1 Report

Dear Authors,

Thank you very much for your answers. There is still a correction to be done before accepting your manuscript for publication.

CT26 instead of C26. Thanks